# Controllable Nano-Crystallization in Fluoroborosilicate Glass Ceramics for Broadband Visible Photoluminescence

**DOI:** 10.3390/nano15020144

**Published:** 2025-01-20

**Authors:** Yuanhang Xiang, Yi Long, Peiying Cen, Sirang Liu, Zaijin Fang, Renjie Jiao

**Affiliations:** 1Sino-French Hoffmann Institute, School of Basic Medical Science, Guangzhou Medical University, Guangzhou 511436, China; xyh890219@gzhmu.edu.cn (Y.X.); 2021118062@stu.gzhmu.edu.cn (S.L.); 2Guangdong Provincial Key Laboratory of Optical Fiber Sensing and Communications, Institute of Photonics Technology, Jinan University, Guangzhou 510632, China; longyi9604@stu2019.jnu.edu.cn; 3College of Physics & Optoelectronic Engineering, Jinan University, Guangzhou 510632, China; 4School of Pharmaceutical Sciences, Guangzhou Medical University, Guangzhou 511436, China; peiyc1007@stu.gzhmu.edu.cn; 5The Second Affiliated Hospital of Guangzhou Medical University, Guangzhou 510261, China

**Keywords:** glass ceramic, nano-crystallization, upconversion luminescence

## Abstract

A transparent fluoroborosilicate glass ceramic was designed for the controllable precipitation of fluoride nanocrystals and to greatly enhance the photoluminescence of active ions. Through the introduction of B_2_O_3_ into fluorosilicate glass, the melting temperature was decreased from 1400 to 1050 °C, and the abnormal crystallization in the fabrication process of fluorosilicate glass was avoided. More importantly, the controlled crystallizations of KZnF_3_ and KYb_3_F_10_ in fluoroborosilicate glass ceramics enhanced the emission of Mn^2+^ and Mn^2+^–Yb^3+^ dimers by 6.7 and 54 times, respectively. Moreover, the upconversion emission color of glass ceramic could be modulated from yellow to white and blue by adjusting the Yb^3+^ concentration. The well-designed glass ceramic is a novel and significant compound to simultaneously provide efficiently coordinated sites for transition metal and rare earth ions. More importantly, the design strategy opens a new way for engineering high-quality oxy-fluoride glass ceramics with properties of excellent stability, controllable nano-crystallization and high-efficiency photoluminescence.

## 1. Introduction

Oxy-fluoride glass is a desired luminescent material that possesses an excellent stability of oxide network structures and a high luminescence efficiency of fluoride environments. In the past decades, extensive research has been conducted on various oxy-fluoride glasses for applications in lighting, solar cells and lasers [1,2,3,4]. Among these, fluorosilicate (FS) glasses have drawn the most attention due to their stable structures of [SiO_4_] frameworks [5,6,7,8,9,10,11]. More importantly, fluoride nanocrystals (NaYF_4_, LaF_3_, YF_3_, SrF_2_, etc.) have been successfully precipitated from the FS glasses via heat treatments to fabricate nano-crystallized glass ceramics (NGCs) due to the thermally metastable states of the glass [12,13,14,15,16,17,18,19,20,21,22,23,24,25,26]. These fluoride nanocrystals provide low-phonon-energy environments for active rare earth (RE) ions and dramatically enhance the luminescent efficiencies of glasses. However, the fabrication of high-quality FS NGCs, in particular, of bulk NGC samples, remains a grand challenge. On one hand, high-silica FS glass exhibits high melting temperature (>1400 °C), which leads to the drastic volatilization of fluorides and the severe erosion to the crucible. Though the melting temperature is decreased by increasing the content of fluorides in the FS glass, the crystallization of the NGC in low-silica glass is usually uncontrollable. Crystals are already precipitated in precursor glasses (PGs) with large sizes due to the low viscosity in networks in low-silica glass. As a result, the optical transmittance of FS–NGC decreases heavily, caused by the scattering of large crystal particles. This has prevented the practical application of FS–NGCs in optical devices. To tackle these questions, it is crucial to design and manufacture novel NGCs characterized by an efficient photoluminescence (PL), a low melting temperature and a controllable precipitation of fluoride nanocrystals.

In this study, a novel fluoroborosilicate (FBS) glass was synthesized by the introduction of B_2_O_3_ into FS glass for controllable crystallizations. The melting temperature of the FBS glass decreased from 1400 to 1050 °C by adding B_2_O_3_, avoiding the severe volatilization of fluorides in glass and reducing the erosion to crucible. The crystalline phases and microstructures of the FBS samples were investigated carefully to compare them with those of the FS samples. Through the heat treatments, KZnF_3_ and KYb_3_F_10_ nanocrystals were controllably precipitated in the FBS NGCs, which provided excellent fluoride crystal environments for enhancing the luminescence efficiency of transition metal (TM) ions and rare earth (RE) ions. To validate this concept, the luminescent characteristics of Mn^2+^- and Yb^3+^-doped NGCs were investigated. The well-designed glass exhibits large potential in the application of multi-color lighting, 3D information storage and tunable lasers.

## 2. Experimental

### 2.1. Materials and Preparations

In this work, the host PGs were selected with a nominal composition of (in mol%) xSiO_2_–(50–x) B_2_O_3_–25ZnF_2_–25KF (x = 40 and 50). The sample was called FBS–PG and FS–PG, respectively. For TM and RE ion doping, MnO and YbF_3_ were selected as doping sources. All materials were produced by Xinzhen New Material Co., Ltd. (Ganzhou, China). All glasses were fabricated via the melt-quenching technique [27]. The stoichiometric mixture of 30 g reagent grade SiO_2_ (99.99%), B_2_O_3_ (99.99%), ZnF_2_ (99.99%), KF (99.99%), YbF_3_ (99.99%) and MnO (99.99%) were fully mixed and melted in quartz crucible for 30 min (the melting temperatures of FBS–PG and FS–PG were 1050 and 1400 °C, respectively). Moreover, the FS–PG sample was also melted in a platinum rhodium crucible. The molten glass was cast into a precooled brass mold, and another copper plate was used to press the glass sample to obtain PG samples. Subsequently, these PG samples were heated at 500 °C and 520 °C for 5 h to prepare NGCs based on the differential scanning calorimetry (DSC) results shown in Figure 1.

### 2.2. Characterization

The thermal stability of the glass was assessed via the DSC curve obtained using the STA449C Jupiter instrument (Netzsch, Selb, Germany) at a heating rate of 5K min^−1^ in an argon atmosphere. High-resolution transmission electron microscope (HR-TEM), high-angle annular dark field TEM (HAADF-TEM) and element mapping were measured with a Tecnai G2 instrument (FEI, Hillsboro, OR, USA). The working voltage was 200 kV, and the current on the sample was ~50 pA. TEM samples, approximately 40 nm in thickness, were obtained using conventional mechanical polishing followed by ion beam milling techniques (PIPS II, GATAN, Pleasanton, CA, USA). The amorphous and crystal states within the glass samples were characterized by X-ray diffraction (XRD) using a D8 advance X-ray diffractometer (Bruker, Fällanden, Switzerland) with Cu/Ka (λ = 0.1541 nm) radiation. The emission spectrum was characterized by an Edinburgh FLS980 fluorescence spectrometer (Edinburgh Instruments, Livingston, UK), utilizing a 450 W xenon lamp as the excitation source. The upconversion (UC) spectra and lifetime curves of the glass were obtained using the same spectrometer. A 980 nm semiconductor laser diode (LD) served as the excitation source for the UC spectra. All properties of samples were measured at room temperature.

## 3. Results and Discussion

The DSC curves presented in Figure 1a,b both contained a crystallization peak temperature (*T_p_*) and a glass transition temperature (*T_g_*). For the controllable precipitation of nanocrystals in FBS–NGCs, the heat treatment temperatures were set between *T_g_* (403 °C) and *T_p_* (532 °C). Thus, the heat treatment temperature of the FBS–PG was set to 500 and 520 °C to prepare NGCs embedded with fluoride nanocrystals. The difference value (ΔT) between *T_g_* and *T_p_* was used to evaluate the controllability of crystallization in glass. The ΔT of FS and FBS–PG was about 90 and 129 °C, respectively, suggesting that the crystallization in FBS–PG was more controllable compared to that in FS–PG. In the compositions of FBS glass, 10 mol% SiO_2_ was substituted by 10 mol% B_2_O_3_, which actually provided 20 mol% [BO_4_] units and thus strengthened the framework of glass. Hence, the introduction of B_2_O_3_ afforded remarkable prospects for the controllable nano-crystallization of fluoride crystals in FBS–NGCs.

In the past investigations, the FS–PG (50SiO_2_-25ZnF_2_-25KF) was proved to be a host for the crystallization of KZnF_3_ crystals to greatly enhance the PL emission of TM ions [28]. However, the PG melted by a platinum rhodium crucible possessed large tendency of crystallization during the glass fabrication process because superabundant fluorides were introduced into the glass. As shown in the inset of Figure 2a, the FS–PG was completely opaque and looked like a ceramic sample. As shown in Figure 2a, the transmittance of FS–PG was sharply reduced to 0% below 600 nm, and the visible light was hardly able to penetrate the sample due to the severe scattering by crystal particles in the PG sample. In fact, a substantial quantity of KZnF_3_ crystals have already precipitated in the PG during the glass fabrication process as proved by the bottom XRD pattern in Figure 2b. Generally, the controllable nano-crystallization in NGC should be achieved by the subsequent heat treatment of PG at a low temperature (a little higher than *T_g_*). The crystallization during glass fabrication process is commonly uncontrollable. Moreover, the average size and the crystalline volume fraction of the KZnF_3_ crystals in the FS–PG was as large as 37.94 nm and 19%, respectively, as calculated from the XRD pattern. This severe and uncontrollable crystallization in the FS–PG makes it difficult to apply in practical optical devices. Moreover, the XRD pattern of the FS–NGC melted by a quartz crucible is also shown in Figure 2b. No peaks corresponding to KZnF_3_ crystals were detected; however, the sharp peaks in the XRD pattern were consistent with the diffraction peaks of K_2_SiF_6_ crystals (No: 85–1382), indicating that the crystalline phase in the NGC changed to K_2_SiF_6_. According to the works by Lin et al. [29], KZnF_3_ crystals tend to precipitate in low-silicon glasses, whereas K_2_SiF_6_ crystals are usually crystallized in the FS glasses containing high concentrations of SiO_2_. The precipitation of K_2_SiF_6_ crystals in the FS–NGC indicates that SiO_2_ from the quartz crucible was dissolved into the glass melt at the high temperature. These results indicate that it is difficult to prepare high-quality NGC based on the FS–PG with the controllable crystallization of KZnF_3_ nanocrystals.

Though the introduction of 10 mol% B_2_O_3_ into the glass to substitute SiO_2_, the total content of fluorides maintained in 50 mol%, and the melting temperature of glass decreased from 1400 to 1050 °C, which reduced the volatilization of fluorides and avoided the severe erosion of the crucible. The FBS–PG exhibited transparent as presented in the inset of Figure 2a. The FBS–PG possessed high transmittance (~76.5% at 600 nm) (Figure 2a) with a thickness of 2 mm. Thus, the uncontrollable crystallization during the fabrication process of glass could be avoided in the FBS–PG.

More importantly, the crystallization in the FBS–NGC was controllable by the subsequent heat treatments at 500 °C. As shown in Figure 3a, a broad band was found in the XRD pattern of PG ascribed to the amorphous phase of glass, suggesting the absence of crystal precipitation during the glass fabrication process. On the contrary, the sharp peaks corresponding to the diffraction peaks of the KZnF_3_ crystals (No: 89–4110) were found in the XRD pattern of NGC. The mainly intense peaks at 2θ = 21.9°, 31.2°, 44.7°, 55.5° and 65.0° were attributed to the diffraction of the (100), (110), (200), (211) and (220) crystal facets for cubic KZnF_3_ crystals, respectively. Moreover, no other diffraction peak was observed in the XRD pattern, proving that only KZnF_3_ crystals were precipitated in the FBS–NGC. The average size of KZnF_3_ nanocrystals was approximately 18.3 nm calculated by Scherrer’s equation, based on the XRD diffraction peak at around 44.6°.

Moreover, the FBS–NGC still possessed high transmittance after the precipitation of KZnF_3_ crystals as shown in Figure 2a. The FBS–NGC exhibited a high transmittance to 71.5% at 600 nm. As the heating temperature rose from 500 to 520 °C, the intensities of all diffraction peaks enhanced, and more KZnF_3_ crystals were precipitated from the glass matrix. These findings verify that the FBS–PG served as an outstanding host for the crystallization of KZnF_3_ and the crystals were controllably precipitated in the FBS–NGC through the heat treatments.

The HAADF-TEM image in Figure 3b illustrates the in situ precipitation of nanocrystals within the glass matrix, with particle sizes ranging from 5 to 30 nm. The TEM elemental mapping patterns (Figure 3d,e) indicate that the distribution of the F and Mn elements correlated closely with that of the nanocrystal particles (Figure 2b). Conversely, the distribution profile of the Si elements was contrary to that of the nanocrystal particles (Figure 3c). These findings unambiguously demonstrate that Mn^2+^ ions were successfully incorporated into the KZnF_3_ nanocrystals, while the K_2_SiF_6_ crystal failed to precipitate in the NGCs. The KZnF_3_ crystal was attributed to a typical cubic perovskite structure with a *Pm-3m* space group, consisting of a 3D network of angle-sharing ZnF_6_ octahedra, where Zn^2+^ ions were situated at the centers of the octahedra [30]. Mn^2+^ exhibited a high octahedral preference energy and possessed the same valence state and comparable ionic radius as Zn^2+^ (R: Zn^2+^ = 0.074 nm, R: Mn^2+^ = 0.067 nm). It was likely to incorporate into the KZnF3 nanocrystal by the substitution for Zn^2+^. Thus, with the precipitation of KZnF_3_ nanocrystals in the FBS–NGCs, Mn^2+^ ions could be confined within fluoride crystal structures, thereby offering excellent environments for high-efficiency luminescence.

The excitation spectra of FBS–PG and FBS–NGC are presented in Figure 4a. In the excitation spectrum of the PG sample, two prominent bands were observed at approximately 352 and 414 nm, corresponding to the electronic transitions of Mn^2+^ from ^6^A_1g_ (S) to the ^4^T_2g_ (D) and ^4^T_2g_ (G) energy levels, respectively. In the PLE spectrum of the GC sample, aside from the broad excitation bands around 352 and 414 nm, two sharp excitation peaks at 330 and 397 nm were also observable. These peaks were attributed to the transitions of Mn^2+^: ^6^A_1g_ (S) → ^4^E_g_ (D) and ^4^E_g_ (G). The PL spectra and emission decay curves of FBS–PG and NGCs doped with 1.0Mn^2+^ are shown in Figure 4b,c. Under 414 nm excitation, the PG exhibited a broadband emission ranging from 500 to 800 nm, as presented in Figure 4a. The emission peak was centered at 640 nm with a full width at half maximum (FWHM) of 127 nm. This emission was attributed to the electronic transition of Mn^2+^ from the ^4^T_1g_ (G) to ^6^A_1g_ (S) energy levels [31]. In the NGC samples, the emission peaks shifted to 588 nm, and the FWHMs narrowed to approximately 66 nm. Given the sensitivity of Mn^2+^ d-d electronic transitions to coordinated environments, the differences in PL emission spectra between PG and NGCs suggest that the coordinated environments of Mn^2+^ ions have been altered during the heat treatments. Compared with the glass matrix, the fluoride crystal exhibited lower phonon energy, which decreased the probability of non-radiative transitions, thereby enhancing the efficiency of Mn^2+^ luminescence. From Figure 4a, it can be further observed that the emission intensity of Mn^2+^ in NGC increased 3.3 and 6.7 times relative to PG after the heat treatments at 500 and 520 °C, respectively. Additionally, the lifetime of Mn^2+^ emission extended from 7.9 to 20.8 ms after the heat treatments. These findings conclusively demonstrate the incorporation of Mn^2+^ ions into the fluoride nanocrystals within the NGCs.

Furthermore, the crystalline phase of Mn^2+^–Yb^3+^ co-doped FBS–PG and NGCs were also investigated and are shown in Figure 5. KZnF_3_ nanocrystals were also precipitated in the Mn^2+^–Yb^3+^ co-doped NGCs as depicted in Figure 5a. Furthermore, the XRD patterns exhibited diffraction peaks at 27° and 53° corresponding to KYb_3_F_10_ crystals. Additionally, the intensities of these diffraction peaks for KYb_3_F_10_ crystals all increased monotonically when the doping concentration of Yb^3+^ rose from 0 to 1.0 mol%. Generally, Yb^3+^ works as a center for crystallization due to its big ionic radius and large potential [32]. Thus, the doping of Yb^3+^ induced the controllable precipitation of KYb_3_F_10_ nanocrystals in the FBS–NGCs. More interestingly, the TEM mapping patterns, depicted in Figure 5b–e, demonstrate that the distributions of Mn and Yb in the co-doped NGC nearly coincided and correlated well with those of F and the crystal particles. It can be assumed that the crystallization of KZnF_3_ in the co-doped NGC was around Yb^3+^, which has been commonly used as a nucleating agent and a crystallization center in NGCs in past investigations [33]. These elemental distributions in the NGC resulted in extremely short distances between RE and TM ions, which was conducive to the formation of Mn^2+^–Yb^3+^ dimers, and F promoted the energy transfer between Yb^3+^ and Mn^2+^.

Figure 6a shows the PL emission spectra of 0.6Mn^2+^–0.2Yb^3+^ co-doped FBS–PG and NGCs under the excitation of a 980 nm LD. Broadband emission around 600 nm was observed in the amorphous PG sample, while the centers of the emission bands in the NGC samples both blue-shifted to 588 nm, which were all ascribed to the UC luminescence of the Mn^2+^–Yb^3+^ dimers [34,35]. In NGC samples, KZnF_3_ and KYb_3_F_10_ crystals offered a low-phonon-energy crystal environment for Mn^2+^ and Yb^3+^, respectively. This not only decreased non-radiative transition probabilities but also shortened the distance between Mn^2+^ and Yb^3+^, facilitating the formation of Mn^2+^–Yb^3+^ dimers. Consequently, as illustrated in Figure 6a, when the samples were heated at 500 and 520 °C for 5 h, respectively, the UC emission of the Mn^2+^–Yb^3+^ dimers in NGC enhanced 45 and 54 times compared to PG. Furthermore, the emission lifetime of the Mn^2+^–Yb^3+^ dimers extended from 1.1 to 8.4 ms following the heat treatments, as presented in Figure 6b. To further investigate the luminescence mechanism of Mn^2+^–Yb^3+^, the dependence of UC emission on the excitation power of a 980 nm laser was examined. The double-logarithmic plots depicting the excitation power dependence on the 588 nm emission intensities, presented in Figure 6c, indicate that the broadband emission around 588 nm in the NGC arose from a two-photon process associated with the UC luminescence of Mn^2+^–Yb^3+^ dimers. Additionally, the UC emission spectra of 1.0Yb^3+^-doped PG and GCs, excited by a 980 nm LD, revealed intense emission peaks around 480 nm in the GCs’ spectra, shown in Figure 6d, which were attributed to the UC emission of Yb^3+^–Yb^3+^ pairs [36,37]. Owing to the KYb_3_F_10_ nanocrystals’ precipitation in NGCs, Yb^3+^ ions were confined in fluoride crystals with short ionic distances, which promoted the Yb^3+^–Yb^3+^ pairs formation. The UC emission intensities of Yb^3+^ in the NGCs were both higher as compared to the PG, as shown in Figure 6d. Because more KYb_3_F_10_ crystals were precipitated in the NGC heated at 520 °C, the ionic distances between Yb^3+^ were shorter. The UC emission intensity of Yb^3+^ in the NGC heated at 520 °C was lower than that at 500 °C due to the concentration-quenching effect. Therefore, enhanced UC emissions of Mn^2+^–Yb^3+^ dimers and Yb^3+^–Yb^3+^ pairs could both be achieved in the FBS–NGCs because of the controllable nano-crystallization of KZnF_3_ and KYb_3_F_10_ crystals.

As mentioned above, GCs exhibited both yellow UC emission around 588 nm and blue UC emission near 480 nm. According to the blue-yellow two-primary-color principle, white emission can be obtained by the combination of the blue and yellow emissions [38]. As shown in Figure 7a, in the spectrum of 1.0Yb^3+^ single-doped NGC, an intense blue emission at 480 nm was detected, which was ascribed to the UC emission of the Yb^3+^–Yb^3+^ pairs. Yellow emissions of the Mn^2+^–Yb^3+^ dimers around 588 nm appeared in the spectra when Mn^2+^ was co-doped in the NGCs. The yellow emission intensity enhanced with the increasing Mn^2+^ doping concentration. When the Mn^2+^ concentration was increased to 0.6 mol% and the Yb^3+^ concentration was reduced to 0.2 mol%, the spectrum predominantly featured the yellow emission of Mn^2+^–Yb^3+^ dimers. The chromaticity coordinate diagram corresponding to the emission spectra in Figure 7a is depicted in Figure 7b. Upon excitation with a 980 nm LD, the Yb^3+^ single-doped NGC emitted pure blue light. White emissions could be observed in the 0.1Mn^2+^–1.0Yb^3+^ and 0.05Mn^2+^–1.0Yb^3+^ NGCs. When the Mn^2+^ content was further increased and the Yb^3+^ concentration was low, the NGC emitted pure yellow light. Hence, the emission color of the NGC could be nearly continuously modulated from blue to white and yellow via adjusting the doping concentrations of Mn^2+^ and Yb^3+^. The FBS–NGCs provide significant compounds for the promising applications in white/multi-color lighting and tunable fiber lasers.

## 4. Conclusions

In summary, to achieve efficient and tunable luminescence, a high-quality FBS–NGC was designed and fabricated. The introduction of B_2_O_3_ into the glass resulted in a reduction of the melting temperature from 1400 to 1050 °C. The designed FBS–NGCs were more stable and possessed higher transmittance than the FS–PG. Meanwhile, KZnF_3_ nanocrystals were controllably precipitated within the Mn^2+^-doped FBS–NGCs, resulting in a 6.7-fold enhancement of Mn^2+^ luminescence. Moreover, KZnF_3_ and KYb_3_F_10_ nanocrystals were simultaneously precipitated in the co-doped NGCs, which confined Mn^2+^ and Yb^3+^ in fluoride crystals, respectively. In the NGCs, the UC emission intensity of Mn^2+^–Yb^3+^ dimers was enhanced 54-fold compared to that in the PG, and the UC emission color was modulated from blue to white and yellow via adjusting the doping concentrations. The well-designed NGCs provide significant optical gain materials for applications in white LEDs, 3D information storages, solar cells and tunable fiber lasers. More importantly, the material design strategy developed herein offers a novel pathway for exploiting a diverse array of high-quality oxy-fluoride NGCs, characterized by high transmittance, highly efficient PL emission and controllable nano-crystallization.

## Figures and Tables

**Figure 1 nanomaterials-15-00144-f001:**
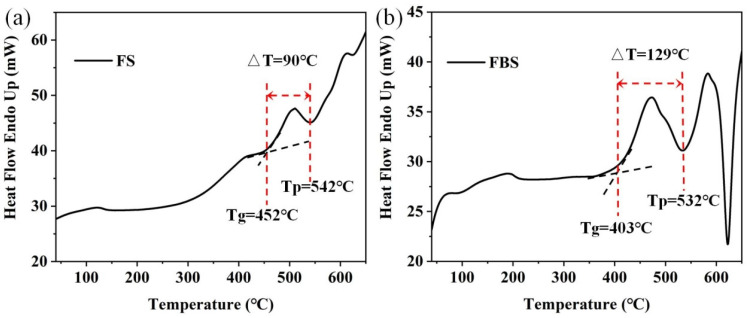
DSC curves of 1.0Mn^2+^-doped (**a**) FS and (**b**) FBS–PGs.

**Figure 2 nanomaterials-15-00144-f002:**
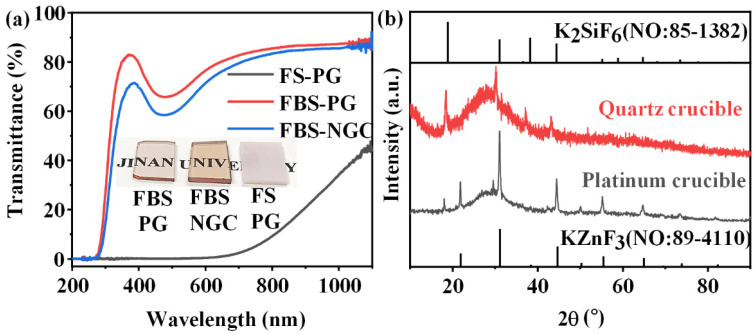
(**a**) Transmission spectra of 1.0Mn^2+^-doped FS–PG, FBS–PG and NGC (treated at 500 °C for 5 h) samples; inset: digital photo of FBS–PG, FBS–NGC and FS–PG samples. (**b**) XRD patterns of FS–PGs melted in different crucibles.

**Figure 3 nanomaterials-15-00144-f003:**
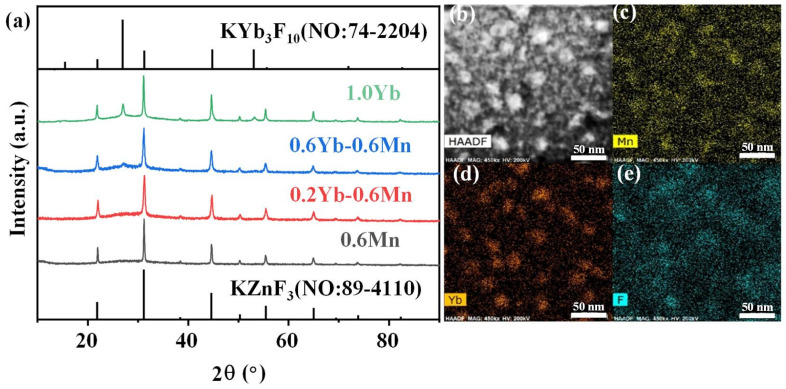
(**a**) XRD patterns of 1.0Mn^2+^-doped FBS–PG and GCs (treated at 500 and 520 °C for 5 h). (**b**) A typical HAADF-TEM image of the 1.0Mn^2+^-doped FBS–GC. (**c**–**e**) TEM elemental mapping images for Yb, F and Mn corresponding to region in (**b**) with relative concentrations indicated by color intensity.

**Figure 4 nanomaterials-15-00144-f004:**
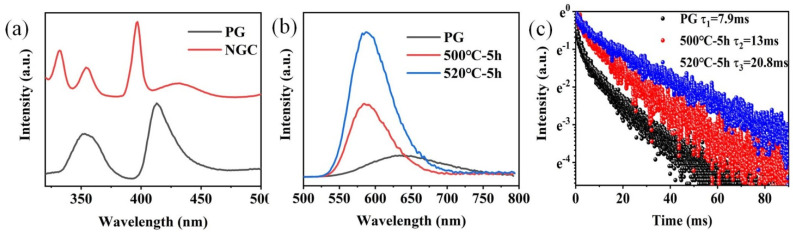
(**a**) Excitation spectra of 1.0Mn^2+^-doped FBS–PG (emission at 620 nm) and NGC (emission at 588 nm). (**b**) PL spectra of 1.0Mn^2+^-doped FBS–PG and NGCs (incubated at 500 and 520 °C for 5 h). (**c**) Emission decay curves of 1.0Mn^2+^-doped FBS–PG (at 640 nm emission) and NGCs (at 588 nm emission).

**Figure 5 nanomaterials-15-00144-f005:**
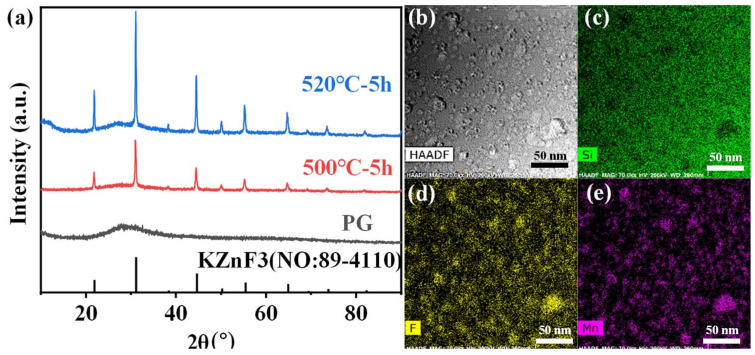
(**a**) XRD patterns of Mn^2+^ single-doped, Mn^2+^–Yb^3+^ co-doped and Yb^3+^ single-doped FBS–GCs (treated at T ~500 °C for 5 h). (**b**) A typical HAADF-TEM image of the 1.0Mn^2+^–1.0Yb^3+^ co-doped FBS–GC. (**c**–**e**) TEM mapping for Mn, Si and F elements, respectively, corresponding to (**b**).

**Figure 6 nanomaterials-15-00144-f006:**
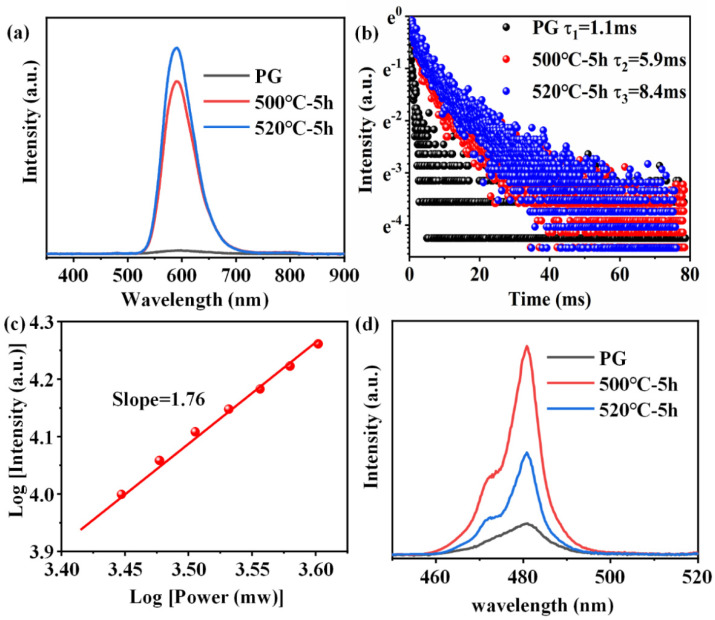
(**a**) Emission spectra of 0.6Mn^2+^–0.2Yb^3+^ co-doped FBS–PG and NGCs excited by a 980 nm LD. (**b**) Decay curves of emissions for Mn^2+^ in FBS–PG and GCs. (**c**) Double-logarithmic plots of the excitation power dependency on the 588 nm emission intensity of 0.6Mn^2+^–0.2Yb^3+^ co-doped FBS–NGC. (**d**) Emission spectra of 1.0Yb^3+^-doped PG and GCs.

**Figure 7 nanomaterials-15-00144-f007:**
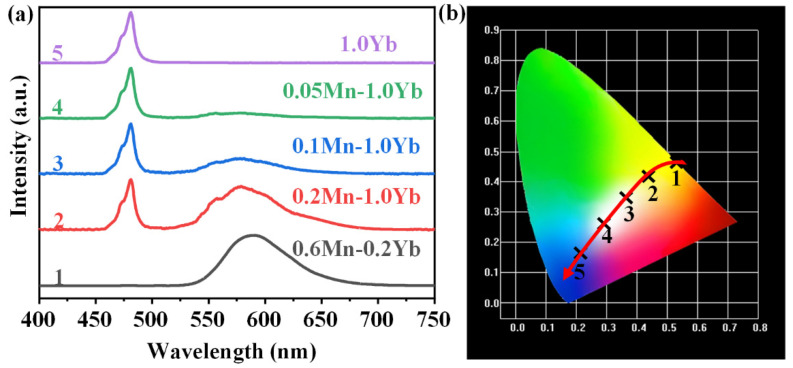
(**a**) The UC emission spectra of FBS–NGCs (heat treated at 500 °C) doped with different concentrations of Mn^2+^ and Yb^3+^ under an excitation of 980 nm LD. (**b**) CIE chromaticity coordinates corresponding to the samples in (**a**).

## Data Availability

Data are contained within the article.

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
