# Peer review of "Controllable Nano-Crystallization in Fluoroborosilicate Glass Ceramics for Broadband Visible Photoluminescence"

_nanomaterials, 2025, doi:10.3390/nano15020144_

Round 1

Reviewer 1 Report

Comments and Suggestions for Authors

Dear Editor

The manuscript entitled "Controllable nano-crystallization in fluoroborosilicate glass-ceramics for broadband visible photoluminescence" presents results on synthesis and luminescence of glass-ceramics with broadband emission in the vis range. 

Manuscript can be consider for publication after several corrections that should be answered by authors:

Abstract, introduction and experimental sections are OK

About Results and Discussion:

Fig.1(a and b). DTA curves should include data from room temperature.

Fig.1b does not seem to clearly reveal crystallization peak temperature and a glass transition temperature. Authors should consider to repeat with slow speed (5K/min) in order to confirm the presented data.

Fig2(a) This figure is not explicit named in the text. Authors can rewrite corresponding paragraph when transmittances are explained. 

About the uncontrolled crystallization, is it not possible to use platinum crucible in order to avoid the undesired presence of K2SiF6 crystals?

Authors do not explain why 520 ºC is the maximum temperature to optimize the crystallization. Perhaps with new DTA data, this value could be extended. Please consider it.

About XRD data of FBS nGC, authors do not estimated their sizes, to compare with the 5-30 nm range, obtained from only one image of HAADF. Please consider. If not, some comments about non homogeneity of samples would be welcome.

About luminescence of PBS nGC, authors could include excitation spectra of glass and glass-ceramics, in supporting information, to justify the excitation wavelengths used in each case. 

About nGCs comprising KYb3F10 nanocrystals, authors could obtain from different nanocrystals, the elemental composition to confirm if nanocrystals are KZn3F10 /Yb.Mn doped) or KYb3F10. Please consider it.

Can authors explain why select 0.6Mn-0.2Yb and not 0.6Mn-0.6Yb co-doped samples for luminescence measurements?

About luminescence of Yb single doped nGC, authors can confirm the origin of 480 nm peak with a long pass filter (F680 or higher) near the laser diode output, to avoid false signals. Just to confirm the origin of observed luminescence. Another possible origin of blue 480 nm emission can be Tm3+ traces, which can be discarted if 650 nm Tm emission is not presented. Please, confirm it.

Author Response

Comments from Reviewer 1:

The manuscript entitled "Controllable nano-crystallization in fluoroborosilicate glass-ceramics for broadband visible photoluminescence" presents results on synthesis and luminescence of glass-ceramics with broadband emission in the vis range. Manuscript can be consider for publication after several corrections that should be answered by authors:

Abstract, introduction and experimental sections are OK

About Results and Discussion:

Response: Thank you very much for your positive comments on our manuscript. We also appreciate your time and effort to help us improve the quality of our manuscript. Now according to your requirements, we offer a point-to-point response to address the involved issues. Please find them from the below response.

  1. Fig.1(a and b). DTA curves should include data from room temperature.

Response: Thank you very much for your suggestion. New DSC curves starting from room temperature are presented in Fig. 1 in the revised manuscript.

  1. Fig.1b does not seem to clearly reveal crystallization peak temperature and a glass transition temperature. Authors should consider to repeat with slow speed (5K/min) in order to confirm the presented data.

Response: Thank you very much for your suggestion. New DSC curves were measured with speed of 5K/min according to your suggestions. The DSC curves are presented in Fig. 1 and the crystallization peak temperature and glass transition temperature are shown in the curves.

  1. Fig2(a) This figure is not explicit named in the text. Authors can rewrite corresponding paragraph when transmittances are explained.

Response: Thank you very much for your suggestion. The missed descriptions have been added in revised manuscript.

  1. About the uncontrolled crystallization, is it not possible to use platinum crucible in order to avoid the undesired presence of K2SiF6 crystals?

Response: Thank you very much for your question. For the FS glass ceramic, if glass was prepared in platinum crucible, crystallization of KZnF3 crystals starts from glass fabrication process. This crystallization is uncontrollable, the crystal sizes are large and transmittance of sample is low, which is bad for the optical applications. When the glass was prepared in quartz crucible, the glass melt reacts with quartz crucible easily at high preparing temperature (~ 1450 oC), a large number Si dissolves into the glass. The crystallization during glass fabrication process is avoided. After heat treat, KZnF3 is not precipitated but K2SiF6 crystals are observed in the glass ceramic. K2SiF6 crystals make no contribution for the enhancement of luminescence of Mn2+.

For FBS glass ceramic, the glass is prepared in quartz crucible at 1050 oC. Owing the adding of B-O networks, fluoride networks are more dispersed, no KZnF3 crystal is precipitated during glass fabrication process. After heat treatments, KZnF3 crystals are controllable precipitated in glass ceramics. More importantly, no K2SiF6 crystal is observed in FBS glass ceramics due to the decrease of reaction between glass melt and quartz crucible at lower preparing temperature (1050 oC).

  1. Authors do not explain why 520 ºC is the maximum temperature to optimize the crystallization. Perhaps with new DTA data, this value could be extended. Please consider it.

Response: Thank you very much for your suggestion. When heat treatment temperature exceeds 520 oC, the emission intensity of Mn2+ is enhanced, while the transmittance of glass ceramic decreases obviously. Therefore, we chose 520 oC as the maximum heat treatment temperature. The relative description is added in the revised manuscript.

  1. About XRD data of FBS nGC, authors do not estimated their sizes, to compare with the 5-30 nm range, obtained from only one image of HAADF. Please consider. If not, some comments about non homogeneity of samples would be welcome.

Response: Thank you very much for your suggestion. The XRD diffraction peak around 44.6o was selected for the calculation of average size of KZnF3 nanocrystals. It was calculated to be approximately 18.3 nm. The crystals are uniformly distributed in the glass ceramics. The relative description is added in the revised manuscript.

  1. About luminescence of PBS nGC, authors could include excitation spectra of glass and glass-ceramics, in supporting information, to justify the excitation wavelengths used in each case.

Response: Thank you very much for your suggestion. The excitation spectra of Mn2+ doped glass and NGC are shown in Fig. 4a in the revised manuscript. The relative descriptions about excitation spectra have been added in the revised manuscript.

  1. About nGCs comprising KYb3F10 nanocrystals, authors could obtain from different nanocrystals, the elemental composition to confirm if nanocrystals are KZn3F10 /Yb.Mn doped) or KYb3F10. Please consider it.

Response: Thank you very much for your suggestion. We have tried to determine the elemental composition of different crystals by EDS measurements. However, the two crystals are distributed very closely as shown in Fig. 5(b)-(d). It is difficult to distinguish the two crystals.

  1. Can authors explain why select 0.6Mn-0.2Yb and not 0.6Mn-0.6Yb co-doped samples for luminescence measurements?

Response: Thank you very much for your question. We have measure the emission spectra of 0.6Mn-xYb (x=0.1-0.6) doped NGC, the emission intensity of 0.6Mn-0.2Yb NGC is the highest. The emission intensity of Mn2+ around 588 nm decreases when Yb3+ concentration is increased to 0.6%.

  1. About luminescence of Yb single doped nGC, authors can confirm the origin of 480 nm peak with a long pass filter (F680 or higher) near the laser diode output, to avoid false signals. Just to confirm the origin of observed luminescence. Another possible origin of blue 480 nm emission can be Tm3+ traces, which can be discarted if 650 nm Tm emission is not presented. Please, confirm it.

Response: Thank you very much for your suggestion. We have re-measured the emission spectra according to your suggestion. By using a 720 nm long pass near the laser diode, the emission peaks around 480 nm are also observed, and no emission peak around 650 nm is observed.

Reviewer 2 Report

Comments and Suggestions for Authors

This work investigates some fluoroborosilicate glasses on photoluminescence properties. Below are my comments, and my recommendation is a major revision.

 1.       In DTA curves, how do you determine Tg and Tp temperatures? Tp of FS is obvious, but others are not clear.

 2.       Eventually, how are actual chemical compositions? Especially, does F remain same as the nominal amount?

 3.       In Fig. 2 (a), what origins ado you consider about absorption bands at ~500 nm?

 4.       In TEM images, please insert a typical spatial scale. It may be shown in the figure, I can’t see it since they may be too small.

 5.       In Figure 4, what physical origins do you consider about the emission from Mn2+? I mean, are they from Mn in crystal phases or in glass phase? Simply considering, emission in PG may be from glass and other from crystal.

 6.       In PL decay, why are decay times so different in spite of similar emission wavelength? Generally, the decay time and emission wavelength are in proportional when we assume the same electron transitions.

 7.       In figure 5, why do spectral shapes change by addition of Yb3+? Generally, Yb3+ do not show absorption on those wavelengths, and emission due to charge transfer does not appear in glasses at room temperature. Further, main emission of Yb3+ appears at ~1030 nm, and it should be observed. Furthermore, “the energy transfer between Yb3+ and Mn2+” is unclear. I require detailed explanation.

 8.       How do you confirm UC? Generally, in such a situation (excitation at longer wavelength), two mechanisms are possible. One is UC and the other is optically stimulated luminescence. I cannot understand from the current manuscript how you conclude the observed emission is UC.

Author Response

Comments from Reviewer 2:

This work investigates some fluoroborosilicate glasses on photoluminescence properties. Below are my comments, and my recommendation is a major revision.

Response: Thank you very much for your positive comments on our manuscript. We also appreciate your time and effort to help us improve the quality of our manuscript. Now according to your requirements, we offer a point-to-point response to address the involved issues. Please find them from the below response.

  1. In DTA curves, how do you determine Tg and Tp temperatures? Tp of FS is obvious, but others are not clear.

Response: Thank you very much for your suggestion. Crystals are easier precipitated in FS NGC, the temperatures of crystallization peak (Tp) for FS glass is obvious. New DSC curves were measured with slower speed of 5K/min. The DSC curves starting from room temperature are presented in Fig. 1 in the revised manuscript. The Tp of FBS glass is more obvious.

  1. Eventually, how are actual chemical compositions? Especially, does F remain same as the nominal amount?

Response: Thank you very much for your suggestion. We have quantitatively established the compositions of the FS and FBS glasses by using X-Ray Fluorescence (XRF) measurements. The measured compositions differ from the nominal values due to the volatilization of fluorine. The fluorine losses for FS and FBS glasses is calculated to 32.54 and 19.49 mass%, respectively (Table R1). The lower preparation temperature of the FBS glass reduces the fluorine loss.

Table R1. Compositions of the FS and FBS glasses

Nominated (mass %)

Measured (mass %)

Fluorine loss (%)

FS glass

20.28

13.68

32.54

FBS glass

20.01

16.11

19.49

  1. In Fig. 2 (a), what origins ado you consider about absorption bands at ~500 nm?

Response: Thank you very much for your question. The absorption bands at ~500 nm are ascribed to the typical transitions of Mn2+ from 6A1 (S) to 4T1 (G) levels.

  1. In TEM images, please insert a typical spatial scale. It may be shown in the figure, I can’t see it since they may be too small.

Response: Thank you very much for your suggestion. The spatial scales are added in the TEM images according to your suggestions (Fig. 3 and Fig. 5).

  1. In Figure 4, what physical origins do you consider about the emission from Mn2+? I mean, are they from Mn in crystal phases or in glass phase? Simply considering, emission in PG may be from glass and other from crystal.

Response: Thank you very much for your question. In the glass, the broadband emission centered at 640 nm is attributed to the electronic transition from 4T1g (G) to 6A1g (S) energy levels of Mn2+ in the glass. In the NGC, the emission centers both shift to 588nm, which mainly are ascribed to the emission of Mn2+ in KZnF3 crystals and are similar to the reported results in KZnF3 nanocrystal materials [1].

  1. In PL decay, why are decay times so different in spite of similar emission wavelength? Generally, the decay time and emission wavelength are in proportional when we assume the same electron transitions.

Response: Thank you very much for your question. In glass, the probability of non-radiative transition is high, the excited energy transfers to the defect of glass networks quickly. The lifetime of Mn2+ emission in glass is short. In NGC, Mn2+ enter KZnF3 crystals, the probability of non-radiative transition in fluoride crystal is low. Thus, the lifetime of Mn2+ emission in NGC is longer. The lifetime increases when heat treatment temperature is increased because more Mn2+ ions enter KZnF3 crystals.

  1. In figure 5, why do spectral shapes change by addition of Yb3+? Generally, Yb3+ do not show absorption on those wavelengths, and emission due to charge transfer does not appear in glasses at room temperature. Further, main emission of Yb3+ appears at ~1030 nm, and it should be observed. Furthermore, “the energy transfer between Yb3+ and Mn2+” is unclear. I require detailed explanation.

Response: Thank you very much for your question. Excited by 980 nm laser diode, emission peak at 480 nm and emission band around 580 nm are observed in Mn-Yb codoped NGCs, which are attributed to the UC emission of Yb3+-Yb3+ pairs and Mn2+-Yb3+ pairs, respectively. When Yb content is low, the emission spectra mainly contain the emission of Mn2+ at 580 nm. When Yb content is high, the emission intensity of Yb3+-Yb3+ pairs at 480 nm increases. The spectral shapes change by addition of Yb3+.

Excited by 980 nm laser diode, the emission of Yb3+ at 1030 nm is observed. However, we measured the spectra from 400 to 750 nm by using a short pass filter (400-800nm) behind the sample. Emission bands around 580 nm are observed, which are attributed to the energy level transitions of Mn2+. These prove that the energy transfers between Yb3+ and Mn2+ exist in our NGC. The detailed mechanism of the energy transfers has been explained in our previous work [2]. Actually, the energy transfers between Yb3+ and Mn2+ have been observed in other materials at room temperature [3]. The references have been added in the revised manuscript.

  1. How do you confirm UC? Generally, in such a situation (excitation at longer wavelength), two mechanisms are possible. One is UC and the other is optically stimulated luminescence. I cannot understand from the current manuscript how you conclude the observed emission is UC.

Response: Thank you very much for your question. Excited by 980 nm laser diode, emission band around 580 nm are observed in Mn-Yb codoped NGCs. The emission is same with that in Mn2+ single-doped NGC excited by 397 nm light. The emission is attributed to transitions of Mn2+ from 4T1(G)→6A1(S) levels. The emission is not optically stimulated luminescence. Moreover, the double-logarithmic plots of the excitation power dependency on the 588 nm emission intensities, presented in Fig. 6(c), reveal that the broadband emission around 588 nm in the NGC is a two-photon process and attributed to the UC luminescence. Additionally, the UC emissions of Yb and Mn have been investigated clearly in our previous works and other papers [1-3].

References

[1] E.H. Song, Z.T. Chen, M. Wu, S. Ding, S. Ye, S.F. Zhou, Q.Y. Zhang, Room‐temperature wavelength‐tunable single‐band upconversion luminescence from Yb3+/Mn2+ Codoped Fluoride Perovskites ABF3, Adv. Opt. Mater. 4 (2016) 798-806.

[2] Y. Long, J. Li, Z. Fang, B. Guan, Modulation of activator distribution by phaseseparation of glass for efficient and tunable upconversion luminescence, RSC Adv., 10 (2020) 12217.

[3] X. Liu, C. Cheng, N. Zeng, X. Li, Q. Jiao, C. Lin, S. Dai, Tunable broadband upconversion luminescence from Yb3+/Mn2+ co-doped dual-phase glass ceramics, Ceram. Int., 46 (2020) 5271-5277.

Round 2

Reviewer 1 Report

Comments and Suggestions for Authors

Dear Editor,

Once the original version of the manuscript have been revised, including partially some of suggestions, in my opinion authors should include two minor aspects, included below, in order to enhance the clarity of explanations along the text. 

If these two aspects are included, it can be considered for publication.

Sincerely yours

1.Authors do not indicate the reason to change DTA by DSC techniques. Different parameters are measured in each case, although equivalent information can be inferred. Please, clarify or at least include a sentence in the revised version of the text, in order to understand the information extracted.

 2. Authors should included their own affirmation: “We have measure the emission spect of 0.6Mn-xYb (x=0.1-0.6) doped NGC, the emission intensity of 0.6Mn-0.2Yb NGC is the highest. The emission intensity of Mn2+ around 588 nm decreases when Yb3+ concentration is increased to 0.6%” in order to support the selection of the studied composition. Please include it

Author Response

1.Authors do not indicate the reason to change DTA by DSC techniques. Different parameters are measured in each case, although equivalent information can be inferred. Please, clarify or at least include a sentence in the revised version of the text, in order to understand the information extracted.

Response: Thank you very much for your question. In order to clearly present the crystallization peaks, we measured the DSC curves. The crystallization peaks in DSC curves is more obvious than those in DTA curves.

  1. Authors should included their own affirmation: “We have measure the emission spect of 0.6Mn-xYb (x=0.1-0.6) doped NGC, the emission intensity of 0.6Mn-0.2Yb NGC is the highest. The emission intensity of Mn2+ around 588 nm decreases when Yb3+ concentration is increased to 0.6%” in order to support the selection of the studied composition. Please include it.

Response: Thank you very much for your suggestion. The emission spectra of 0.6Mn-xYb (x=0.1-0.6) doped NGCs are presented in Fig. R1. The UC emission of Mn2+ (around 588 nm) increases firstly, reaching a maximum at 0.2Yb0.6Mn, and then decreases when the Yb3+ content is further increased. The emission intensity of 0.6Mn-0.2Yb NGC is the highest. Therefore, the luminescence properties of 0.2Yb0.6Mn NGC sample was investigated to study the UC emission of Mn-Yb pairs.

Fig. R1 UC emission spectra of 0.6Mn-xYb (x=0.1-0.6) doped NGCs.

Reviewer 2 Report

Comments and Suggestions for Authors

Regarding to my last comment 8, your response does not make sense. Here, your situation is longer excitation wavelength and shorter emission wavelength. This situation can be possible by UC or optically stimulated luminescence (OSL). Electron transitions of specific ions and broadness of emission band cannot be logical evidences to distinguish these two mechanisms. Generally, to distinguish them, change of emission intensity is checked. In the case of UC, emission intensity does not change during the excitation, but in the case of OSL, the emission intensity changes under stimulation. Please confirm.

Author Response

Regarding to my last comment 8, your response does not make sense. Here, your situation is longer excitation wavelength and shorter emission wavelength. This situation can be possible by UC or optically stimulated luminescence (OSL). Electron transitions of specific ions and broadness of emission band cannot be logical evidences to distinguish these two mechanisms. Generally, to distinguish them, change of emission intensity is checked. In the case of UC, emission intensity does not change during the excitation, but in the case of OSL, the emission intensity changes under stimulation. Please confirm.

Response: Thank you very much for your suggestion. We measured the emission spectra under the excitation of 980 nm laser diode for 30 min. The emission intensity dose not change during the excitation process. We consider that the emission is attributed to UC luminescence. Thank you for teaching us to distinguish OSL and UC luminescence.
